# Causal Regularization

**Dominik Janzing**
Amazon Research Tübingen
Germany
`janzind@amazon.com`

## Abstract

We argue that regularizing terms in standard regression methods not only help against overfitting finite data, but sometimes also help in getting better *causal* models. We first consider a multi-dimensional variable linearly influencing a target variable with some multi-dimensional unobserved common cause, where the confounding effect can be decreased by keeping the penalizing term in Ridge and Lasso regression even in the population limit. The reason is a close analogy between overfitting and confounding observed for our toy model. In the case of overfitting, we can choose regularization constants via cross validation, but here we choose the regularization constant by first estimating the strength of confounding, which yielded reasonable results for simulated and real data. Further, we show a 'causal generalization bound' which states (subject to our particular model of confounding) that the error made by interpreting any non-linear regression as *causal* model can be bounded from above whenever functions are taken from a not too rich class.

## 1   Introduction

Predicting a scalar target variable $Y$ from a $d$-dimensional predictor $\mathbf{X} := (X_1, \ldots, X_d)$ via appropriate regression models is among the classical problems of machine learning [1]. In the standard supervised learning scenario, some finite number of observations, independently drawn from an unknown but fixed joint distribution $P_{Y,\mathbf{X}}$, are used for inferring $Y$-values corresponding to unlabelled $\mathbf{X}$-values. To solve this task, regularization is known to be crucial for obtaining regression models that generalize well from training to test data [2]. Deciding whether such a regression model admits a *causal* interpretation is, however, challenging. Even if causal influence from $Y$ to $\mathbf{X}$ can be excluded (e.g. by time order), the statistical relation between $\mathbf{X}$ and $Y$ cannot necessarily be attributed to the influence of $\mathbf{X}$ on $Y$. Instead, it could be due to possible common causes, also called 'confounders'. For the case where common causes are known and observed, there is a huge number of techniques to infer the causal influence[1], e.g., [3], addressing different challenges, for instance, high dimensional confounders [4] or the case where some variables other than the common causes are observed [5], just to mention a few of them. If common causes are not known, the task of inferring the influence of $\mathbf{X}$ on $Y$ gets incredibly hard. Given observations from any further variables other than $\mathbf{X}$ and $Y$, conditional independences may help to detect or disprove the existence of common causes [5], and so-called instrumental variables may admit the identification of causal influence [6].

Here we consider the case where only observations from $\mathbf{X}$ and $Y$ are given. In this case, naively interpreting the regression model as causal model is a natural baseline. We show that strong regularization increases the chances that the regression model contains some causal truth. We are aware of the risk that this result could be mistaken as a justification to ignore the hardness of the problem and blindly infer causal models by strong regularization. Our goal is, instead, to inspire a

discussion on to what extent causal modelling should regularize even in the infinite sample limit due to some analogies between *generalizing across samples from the same distribution* and *'generalizing' from observational to interventional distributions*, which appear in our models of confounding, while they need not apply to other confounding scenarios. The idea is not entirely novel since it is tightly linked to several ideas that are 'floating around' in the machine learning community for a while. It is believed (and can be proven subject to appropriate model assumptions) that finding statistical models that generalize well across different background conditions is closely linked to finding *causal* models [7, 8, 9, 10].[2] It is then natural to also believe that generalizing across different environment is related to generalizing across different samples. Accordingly, [12] describes regularization techniques for linear regression that help generalizing across certain shift perturbations. Here we describe a scenario for which the analogy between 'regularizing against overfitting' and 'regularizing against confounding' gets as tight as possible in the sense that the same regularization helps for both purposes. Due to this theoretical focus, we prefer to work with the simplest non-trivial scenario rather than looking for the most relevant or most realistic case.

**Scenario 1: inferring a linear statistical model**    To explain the idea, we consider a linear statistical relation between $\mathbf{X}$ and $Y$:

$$Y = \mathbf{X}\mathbf{a} + E, \tag{1}$$

where $\mathbf{a}$ is a column vector in $\mathbb{R}^d$ and $E$ is an uncorrelated unobserved noise variable, i.e., $\Sigma_{\mathbf{X}E} = 0$. Let $\hat{Y}$ denote the column vector of centred renormalized observations $y^i$ of $Y$, i.e., with entries $(y^i - \frac{1}{n}\sum_{i=1}^n y^i)/\sqrt{n-1}$, and similarly, $\hat{E}$ denotes the centred renormalized values of $E$. Likewise, let $\hat{\mathbf{X}}$ denote the $n \times d$ matrix whose $j$-th column contains the centred renormalized observations from $X_j$. Let, further, $\hat{\mathbf{X}}^{-1}$ denote its (Moore-Penrose) pseudoinverse. To avoid overfitting, the least ordinary squares estimator[3]

$$\hat{\mathbf{a}} := \mathrm{argmin}_{\mathbf{a}'} \|\hat{Y} - \hat{\mathbf{X}}\mathbf{a}'\|^2 = \hat{\mathbf{X}}^{-1}\hat{Y} = \mathbf{a} + \hat{\mathbf{X}}^{-1}\hat{E}, \tag{2}$$

is replaced with the Ridge and Lasso estimators

$$\hat{\mathbf{a}}_\lambda^{\mathrm{ridge}} \quad := \quad \mathrm{argmin}_{\mathbf{a}'}\{\lambda\|\mathbf{a}'\|_2^2 + \|\hat{Y} - \hat{\mathbf{X}}\mathbf{a}'\|^2\} \tag{3}$$

$$\hat{\mathbf{a}}_\lambda^{\mathrm{lasso}} \quad := \quad \mathrm{argmin}_{\mathbf{a}'}\{\lambda\|\mathbf{a}'\|_1 + \|\hat{Y} - \hat{\mathbf{X}}\mathbf{a}'\|^2\}, \tag{4}$$

where $\lambda$ is a regularization parameter [13].

So far we have only described the standard scenario of inferring properties of the conditional $P_{Y|X}$ from finite observations $\hat{\mathbf{X}}, \hat{Y}$ without any *causal* semantics.

**Scenario 2: inferring a linear causal model**    We now modify the scenario in three respects. First, we assume that $E$ and $\mathbf{X}$ in (1) correlate due to some unobserved common cause. Second, we interpret (1) in a *causal way* in the sense that setting $\mathbf{X}$ to $\mathbf{x}$ lets $Y$ being distributed according to $\mathbf{x}\mathbf{a} + E$. Using Pearl's do-notation (a crucial concept for formalizing causality) [5], this can be phrased as

$$Y|_{do(\mathbf{X}=\mathbf{x})} = \mathbf{x}\mathbf{a} + E \quad \neq Y|_{\mathbf{X}=\mathbf{x}}, \tag{5}$$

where we don't have equality because $E$ needs to be replaced with $E|_{\mathbf{X}=\mathbf{x}}$ for the observational conditional. Third, we assume the infinite sample limit where $P_{\mathbf{X},Y}$ is known. We still want to infer the vector $\mathbf{a}$ because we are interested in causal statements but regressing $Y$ on $\mathbf{X}$ yields $\hat{\mathbf{a}}$ instead which describes the *observational* conditional on the right hand side of (5).

Conceptually, Scenario 1 and 2 deal with two entirely different problems: inferring $P_{Y|X=x}$ from finite samples $(\hat{\mathbf{X}}, \hat{Y})$ versus inferring the interventional conditional $P_{Y|do(\mathbf{X}=\mathbf{x})}$ from the observational distribution $P_{Y,\mathbf{X}}$. Nevertheless both problems amount to inferring the vector $\mathbf{a}$ and for both scenarios, the error term $\hat{\mathbf{X}}^{-1}\hat{E}$ causes failure of ordinary least squares regression. Only the reason why this term is non-zero differs: in the first scenario it is a finite sample effect, while it results from confounding in the second one. The idea of the present paper is simply that standard regularization

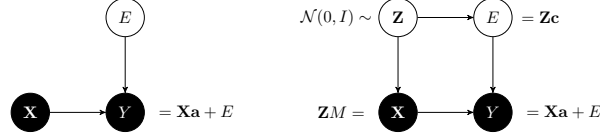

Figure 1: Left: In scenario 1, the empirical correlations between $X$ and $E$ are only finite sample effects. Right: In scenario 2, $\mathbf{X}$ and $E$ are correlated due to their common cause $\mathbf{Z}$. We sample the structural parameters $M$ and $\mathbf{c}$ from distributions in a way that entails a simple analogy between scenario 1 and 2.

techniques do not care about the *origin* of this error term. Therefore, they can temper the impact of confounding in the same way as they help avoiding to overfit finite data.

The paper is structured as follows. Section 2 fleshes out scenarios 1 and 2 in a way that entails that the regression error follows the same distributions. Section 3 proposes a way to determine the regularization parameter in scenario 2 by estimating the strength of confounding via a method proposed by [14]. Section 4 describes some empirical results. Section 5 describes a modified statistical learning theory that states that regression models from not too rich function classes 'generalize' from statistical to causal statements.

## 2 Analogy between overfitting and confounding

The reason why our scenario 2 only considers the *infinite* sample limit of confounding is that mixing finite sample and confounding significantly complicates the theoretical results. The supplement sketches the complications of this case. For a concise description of the population case, we consider the Hilbert space $\mathcal{H}$ of centred random variables (on some probability space without further specification) with finite variance. The inner product is given by the covariance, e.g.,

$$\langle X_i, X_j \rangle := \operatorname{cov}(X_i, X_j). \tag{6}$$

Accordingly, we can interpret $\mathbf{X}$ as an operator[4] $\mathbb{R}^d \to \mathcal{H}$ via $(b_1, \ldots, b_d) \mapsto \sum_j b_j X_j$. Then the population version of (2) reads

$$\tilde{\mathbf{a}} = \operatorname{argmin}_{\mathbf{a}'} \{\|Y - \mathbf{X}\mathbf{a}'\|^2\} = \mathbf{X}^{-1} Y = \mathbf{a} + \mathbf{X}^{-1} E, \tag{7}$$

where the square length is induced by the inner product (6), i.e., it is simply the variance. Extending the previous notation, $\mathbf{X}^{-1}$ now denotes the pseudoinverse of the operator $\mathbf{X}$ [15]. To see that $\mathbf{X}^{-1}E$ is only non-zero when $\mathbf{X}$ and $E$ are correlated it is helpful to rewrite it as

$$\mathbf{X}^{-1} E = \Sigma_{\mathbf{X}\mathbf{X}}^{-1} \Sigma_{\mathbf{X}E}, \tag{8}$$

where we have assumed $\Sigma_{\mathbf{X}\mathbf{X}}$ to be invertible (see supplement for a proof). One can easily show that the empirical covariance matrix $\widehat{\Sigma_{\mathbf{X}E}}$ causing the overfitting error is distributed according to $\mathcal{N}(0, \widehat{\Sigma_{\mathbf{X}\mathbf{X}}} \sigma_E^2 / n)$.[5]

To get the desired analogy between scenarios 1 and 2, we just need a generating model for confounders for which $\Sigma_{\mathbf{X}E}$ is distributed according to $\mathcal{N}(0, \gamma \Sigma_{\mathbf{X}\mathbf{X}})$ for some parameter $\gamma$. The independent source model for confounding described in [14] turned out to satisfy this requirement after some further specification.

**Generating model for scenario 1** The following procedure generates samples according to the DAG in Figure 1, left:

1. Draw $n$ observations from $(X_1, \ldots, X_d)$ independently from $P_X$
2. Draw samples of $E$ independently from $P_E$
3. Draw the vector $\mathbf{a}$ of structure coefficients from some distribution $P_{\mathbf{a}}$
4. Set $\hat{Y} := \hat{\mathbf{X}}\mathbf{a} + \hat{E}$.

**Generating model for scenario 2** To generate random variables according to the DAG in Figure 1, right, we assume that both variables $\mathbf{X}$ and $E$ are generated from the same set of independent sources by applying a random mixing matrix or a random mixing vector, respectively:

Given an $\ell$-dimensional random vector $\mathbf{Z}$ of sources with distribution $\mathcal{N}(0, I)$.

1. Choose an $\ell \times d$ mixing matrix $M$
   and set $\mathbf{X} := \mathbf{Z}M$.
2. Draw $\mathbf{c} \in \mathbb{R}^\ell$ from some distribution $P_{\mathbf{c}}$ and set $E := \mathbf{Z}\mathbf{c}$.
3. Draw the vector $\mathbf{a}$ of structure coefficients from some distribution $P_{\mathbf{a}}$
4. Set $Y := \mathbf{X}\mathbf{a} + E$.

We then obtain:

**Theorem 1** (population and empirical covariances). *Let the number $\ell$ of sources in scenario 2 be equal to the number $n$ of samples in scenario 1 and $P_M$ coincide with the distribution of sample matrices $\hat{\mathbf{X}}$ induced by $P_{\mathbf{X}}$. Let, moreover, $P_{\mathbf{c}}$ in scenario 2 coincide with the distribution of $\hat{E}$ induced by $P_E$ in scenario 1, and $P_{\mathbf{a}}$ be the same in both scenarios. Then the joint distribution of $\mathbf{a}, \Sigma_{\mathbf{XX}}, \Sigma_{\mathbf{XY}}, \Sigma_{\mathbf{XE}}$ in scenario 2 coincides with the joint distribution of $\mathbf{a}, \widehat{\Sigma_{\mathbf{XX}}}, \widehat{\Sigma_{\mathbf{XY}}}, \widehat{\Sigma_{\mathbf{XE}}}$ in scenario 1.*

*Proof.* We have $\widehat{\Sigma_{\mathbf{XX}}} = \hat{\mathbf{X}}^T \hat{\mathbf{X}}$ and $\Sigma_{\mathbf{XX}} = \mathbf{X}^T \mathbf{X} = M^T \mathbf{Z}^T \mathbf{Z} M = M^T M$, where we have used that $\mathbf{Z}$ has full rank due to the uncorrelatedness of its components. Likewise, $\widehat{\Sigma_{\mathbf{XE}}} = \hat{\mathbf{X}}^T \hat{E}$ and $\Sigma_{\mathbf{XE}} = (\mathbf{Z}M)^T \mathbf{Z}\mathbf{c} = M^T \mathbf{c}$. Further, $\widehat{\Sigma_{\mathbf{XY}}} = \hat{\mathbf{X}}^T \hat{\mathbf{X}}\mathbf{a} + \widehat{\Sigma_{\mathbf{XE}}}$ and $\Sigma_{\mathbf{XY}} = \mathbf{X}^T \mathbf{X}\mathbf{a} + \Sigma_{\mathbf{XE}}$. Then the statement follows from the correspondences $M \equiv \hat{\mathbf{X}}, \mathbf{c} \equiv \hat{E}, \mathbf{a} \equiv \mathbf{a}$. $\qquad\square$

Theorem 1 provides a canonical way to transfer any Bayesian approach to inferring $\mathbf{a}$ from $\widehat{\Sigma_{\mathbf{XX}}}, \widehat{\Sigma_{\mathbf{XY}}}$ in scenario 1 to inferring $\mathbf{a}$ from $\Sigma_{\mathbf{XX}}, \Sigma_{\mathbf{XY}}$ in scenario 2. It is known [16], for instance, that (3) and (4) maximize the posterior $p(\mathbf{a}|\hat{\mathbf{X}}, \hat{Y})$ for the priors

$$p_{\text{ridge}}(\mathbf{a}) \quad \sim \quad \exp\left(-\frac{1}{2\tau^2}\|\mathbf{a}\|^2\right) \qquad p_{\text{lasso}}(\mathbf{a}) \sim \exp\left(-\frac{1}{2\tau^2}\|\mathbf{a}\|_1\right), \qquad (9)$$

respectively, if $E \sim \mathcal{N}(0, \sigma_E^2)$ and $\lambda = \sigma_E^2/\tau^2$. Some algebra shows that the only information from $\hat{\mathbf{X}}$ and $\hat{Y}$ that matters is given by $\widehat{\Sigma_{\mathbf{XX}}}$ and $\widehat{\Sigma_{\mathbf{XY}}}$, see supplement. Therefore, (3) and (4) also maximize the posterior $p(\mathbf{a}|\widehat{\Sigma_{\mathbf{XX}}}, \widehat{\Sigma_{\mathbf{XY}}})$ and, employing Theorem 1, the population versions of Ridge and Lasso

$$\tilde{\mathbf{a}}_\lambda^{\text{ridge}} \quad := \quad \operatorname{argmin}_{\mathbf{a}'}\{\lambda\|\mathbf{a}'\|_2^2 + \|Y - \mathbf{X}\mathbf{a}'\|^2\} \qquad (10)$$

$$\tilde{\mathbf{a}}_\lambda^{\text{lasso}} \quad := \quad \operatorname{argmin}_{\mathbf{a}'}\{\lambda\|\mathbf{a}'\|_1 + \|Y - \mathbf{X}\mathbf{a}'\|^2\}, \qquad (11)$$

maximize $p(\mathbf{a}|\Sigma_{\mathbf{XX}}, \Sigma_{\mathbf{XY}})$ after substituting all the priors accordingly.

These population versions, however, make it apparent that we now face the problem that selecting $\lambda$ by cross-validation would be pointless since $\lambda = 0$ had the best cross sample performance. Instead, we would need to know the strength of confounding to choose the optimal $\lambda$.

## 3  Choosing the regularization constant by estimating confounding

The only approaches that directly estimate the strength of confounding[6] from $P_{\mathbf{X}, Y}$ alone we are aware of are given by [19, 14]. The first paper considers only one-dimensional confounders, which is complementary to our confounding scenario, while we will use the approach from the second paper

because it perfectly matches our scenario 2 in Section 2 with fixed $M$. [14] use the slightly stronger assumption that $\mathbf{a}$ and $\mathbf{c}$ are drawn from $\mathcal{N}(0, \sigma_a^2 I)$ and $\mathcal{N}(0, \sigma_c^2 I)$, respectively. We briefly rephrase the method. Using $\tilde{\mathbf{a}}$ in (7) (i.e. the population version of the ordinary least squares solution), they define confounding strength by

$$\beta := \frac{\|\tilde{\mathbf{a}} - \mathbf{a}\|^2}{\|\tilde{\mathbf{a}} - \mathbf{a}\|^2 + \|\mathbf{a}\|^2} \in [0, 1]. \tag{12}$$

It attains 0 iff $\tilde{\mathbf{a}}$ coincides with $\mathbf{a}$ and 1 iff $\mathbf{a} = 0$ and the correlations between $\mathbf{X}$ and $Y$ are entirely caused by confounding.

The idea to estimate $\beta$ is that the unregularized regression vector follows the distribution $\mathcal{N}(0, \sigma_a^2 \mathbf{I} + \sigma_c^2 M^{-1} M^{-T})$. This results from

$$\tilde{\mathbf{a}} = \mathbf{a} + X^{-1} E = \mathbf{a} + M^{-1} \mathbf{c},$$

(see proof of Theorem 1 in [14]). Then the quotient $\sigma_c^2/\sigma_a^2$ can be inferred from the direction of $\hat{\mathbf{a}}$ (intuitively: the more $\hat{\mathbf{a}}$ concentrates in small eigenvalue eigenspaces of $\Sigma_{\mathbf{XX}} = M^T M$, the larger is this quotient). Using some approximations that hold for large $d$, $\beta$ can be estimated from $(\Sigma_{\mathbf{XX}}, \tilde{\mathbf{a}})$. Further, the approximation $\|\tilde{\mathbf{a}} - \mathbf{a}\|^2 + \|\mathbf{a}\|^2 \approx \|\tilde{\mathbf{a}}\|^2$ from [19] yields $\|\mathbf{a}\|^2 \approx (1 - \beta) \cdot \|\tilde{\mathbf{a}}\|^2$. Hence, the length of the true causal regression vector $\mathbf{a}$ can be estimated from the length of $\tilde{\mathbf{a}}$. This way, we can adjust $\lambda$ such that $\|\hat{\mathbf{a}}_\lambda\|$ coincides with the estimated length. Since the estimation is based on a Gaussian (and not a Laplacian) prior for $\mathbf{a}$, it seems more appropriate to combine it with Ridge regression than with Lasso. However, due to known advantages of Lasso[7] (e.g. that sparse solutions yield more interpretable results), we also use Lasso. After all, the qualitative statement that strong confounding amounts to vectors $\hat{\mathbf{a}}$ that tend to concentrate in low eigenvalue subspaces of $\Sigma_{\mathbf{XX}}$ still holds true as long as $\mathbf{c}$ is still chosen from an isotropic prior.

Confounding estimation via the algorithm of [14] requires the problematic decision of whether the variables $X_j$ should be rescaled to variance 1. If different $X_j$ refer to different units, there is no other straightforward choice of the scale. It is not recommended, however, to always normalize $X_j$. If $\Sigma_{\mathbf{XX}}$ is diagonal, for instance, the method would be entirely spoiled by normalization. The difficulty of deciding whether data should be renormalizing beforehand will be inherited by our algorithm.

Our confounder correction algorithm reads:

**Algorithm ConCorr**
1: **Input:** i.i.d. samples from $P(\mathbf{X}, Y)$.
2: Rescale $X_j$ to variance 1 if desired.
3: Compute the empirical covariance matrices $\widehat{\Sigma_{\mathbf{XX}}}$ and $\widehat{\Sigma_{\mathbf{XY}}}$
4: Compute the ordinary least squares regression vector $\hat{\mathbf{a}} := \widehat{\Sigma_{\mathbf{XX}}}^{-1} \widehat{\Sigma_{\mathbf{XY}}}$
5: Compute an estimator $\hat{\beta}$ for the confounding strength $\beta$ via the algorithm in [14] from $\widehat{\Sigma_{\mathbf{XX}}}$ and $\hat{\mathbf{a}}$ and estimate the squared length of $\mathbf{a}$ via

$$\|\mathbf{a}\|^2 \approx (1 - \hat{\beta})\|\hat{\mathbf{a}}\|^2 \tag{13}$$

6: find $\lambda$ such that the squared length of $\hat{\mathbf{a}}_\lambda^{\texttt{ridge/lasso}}$ coincides with the square root of the right hand side of (13).
7: Compute Ridge or Lasso regression model using this value of $\lambda$.
8: **Output:** Regularized regression vectors $\hat{\mathbf{a}}_\lambda^{\texttt{ridge/lasso}}$.

## 4 Experiments

### 4.1 Simulated data

For some fixed values of $d = \ell = 30$, we generate one mixing matrix $M$ in each run by drawing its entries from the standard normal distribution. In each run we generate $n = 1000$ instances of the

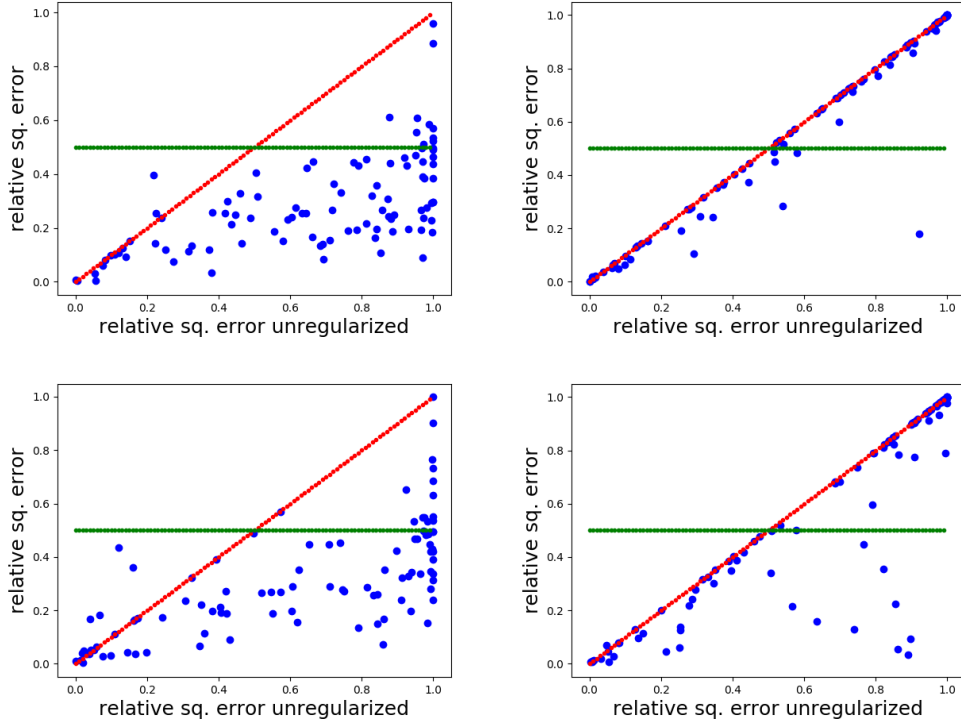

Figure 2: From left to right: RSE versus unregularized RSE (that is, ordinary least square regression) for `Concorr` with Ridge, standard cross-validated Ridge (top, left and right, respectively), and `ConCorr` with Lasso, standard cross-validaded Lasso (bottom, left and right, respectively) for 100 runs (each point representing one run).

$\ell$-dimensional standard normal random vector $\mathbf{Z}$ and compute the $\mathbf{X}$ values by $\mathbf{X} = \mathbf{Z}M$. Afterwards we draw the entries of $\mathbf{c}$ and $\mathbf{a}$ from $\mathcal{N}(0, \sigma_c^2)$ and $\mathcal{N}(0, \sigma_a^2)$, respectively, after choosing $\sigma_a$ and $\sigma_c$ from the uniform distribution on $[0, 1]$. Finally, we compute the values of $Y$ via $Y = \mathbf{X}\mathbf{a} + \mathbf{Z}\mathbf{c} + E$, where $E$ is random noise drawn from $\mathcal{N}(0, \sigma_E^2)$ (the parameter $\sigma_E$ has previously been chosen uniformly at random from $[0, 5]$, which yields quite noisy data). While such a noise term didn't exist in our description of scenario 2, we add it here to also study finite sample effects (without noise, $Y$ depends deterministically on $\mathbf{X}$ for $\ell \leq d$).

To assess whether the output $\hat{\mathbf{a}}_\lambda$ is close to $\mathbf{a}$ we define the relative squared error (RSE) of any regression vector $\mathbf{a}'$ by

$$\epsilon_{\mathbf{a}'} := \frac{\|\mathbf{a}' - \mathbf{a}\|^2}{\|\mathbf{a}' - \mathbf{a}\|^2 + \|\mathbf{a}\|^2} \in [0, 1]$$

This definition is convenient because it yields the confounding strength $\beta$ for the special case where $\mathbf{a}'$ is the ordinary least squares regression vector $\tilde{\mathbf{a}}$.

Figure 2 shows the results. The red and green lines show two different baselines: first, the unregularized error, and second, the error $1/2$ obtained by the trivial regression vector $0$. The goal is to stay below both baselines. Apart from those two trivial baselines, another natural baseline is regularized regression where $\lambda$ is chosen by cross-validation, because this would be the default approach for the unconfounded case. We have used leave-one-out CV from the `Python` package `scikit` for Ridge and Lasso, respectively.

`ConCorr` clearly outperforms cross-validation (for both Ridge and Lasso), which shows that cross-validation regularizes too weakly for causal modelling, as expected. One should add, however, that we increased the number of iterations in the $\lambda$-optimization to get closer to optimal leave-one-out performance since the default parameters of `scikit` already resulted in regularizing more strongly than that (Note that the goal of this paper is not to show that `ConCorr` outperforms other methods.

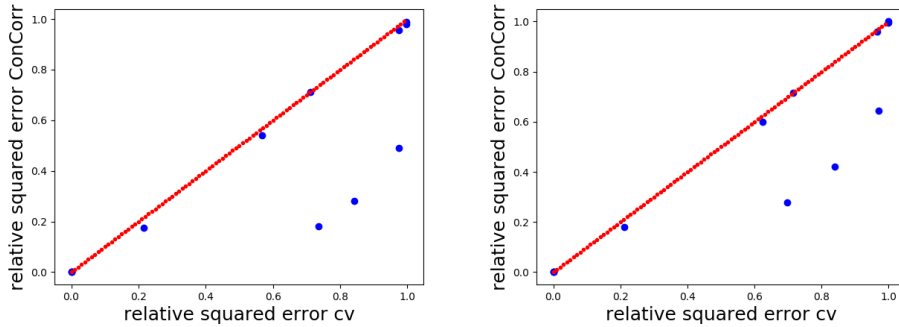

Figure 3: Results for Ridge (left) and Lasso (right) regression for the data from the optical device.

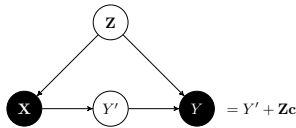

Figure 4: Confounding where $\mathbf{Z}$ influences $Y$ in a linear additive way, while the influence on $\mathbf{X}$ is arbitrary.

Instead, we want to argue that for causal models it is often recommended to regularize more strongly than criteria of *statistical predictability* suggests. If 'early stopping' in common CV algorithms also yields stronger regularization,[8] this can be equally helpful for causal inference, although the way `ConCorr` choses $\lambda$ is less arbitrary than just bounding the number of iterations).

Results for other dimensions were qualitatively comparable if $d$ and $\ell$ were above 10 with slow improvement for larger dimensions, but note that the relevance of simulations should not be overestimated since inferring confounding critically depends on the distribution of eigenvalues of $\Sigma_{\mathbf{XX}}$, which is domain dependent in practical applications.

## 4.2    Real data

In absence of better data sets with known ground truth, we considered two sets used in [14], where ground truth was assumed to be known up to some uncertainty discussed there.

**Optical device**    Here, a Laptop shows an image with extremely low resolution (in their case $3 \times 3$-pixel[9]) captured from a webcam. In front of the screen they mounted a photodiode measuring the light intensity $Y$, which is mainly influenced by the pixel vector $\mathbf{X}$ of the image.

The confounder $W$ is a random voltage controlling two LEDs, one in front of the webcam (and thus influencing $\mathbf{X}$) and the second one in front of the photodiode (thus influencing $Y$). Since $W$ is also measured, the vector $\mathbf{a}_{\mathbf{X},W}$ obtained by regressing $Y$ on $(\mathbf{X}, W)$ is causal (no confounders by construction), if one accepts the linearity assumption. Dropping $W$ yielded significant confounding, with $\beta$ ranging from 0 to 1. We applied `ConCorr` to $\mathbf{X}, Y$ and compared the output with the ground truth. Figure 3, left, show the results for Ridge and Lasso. The $y$-axis is the relative squared error achieved by `ConCorr`, while the $x$-axis is the cross-validated baseline.

The point $(0, 0)$ happened to be met by three cases, where no improvement was possible. One can see that in 3 out of the remaining nine cases (note that the point $(1, 1)$ is also met by two cases), `ConCorr` significantly improved the causal prediction. Fortunately, there is no case where `ConCorr` is worse than the baseline.

**Taste of wine** This data has been extracted from the UCI machine learning repository [22] for the experiments in [14]. The cause $\mathbf{X}$ contains 11 ingredients of different sorts of red wine and $Y$ is the taste assigned by human subjects. Regressing $Y$ on $\mathbf{X}$ yields a regression vector for which the ingredient `alcohol` dominates. Since alcohol strongly correlates with some of the other ingredients, dropping it amounts to significant confounding (assuming that the correlations between alcohol and the other ingredients is due to common causes and not due to the influence of alcohol on the others).

After normalizing the ingredients[10], ConCorr with Ridge and Lasso yielded a relative error of $0.45$ and $0.35$, respectively, while [14] computed the confounding strength $\beta \approx 0.8$, which means that ConCorr significantly corrects for confounding (we confirmed that CV also yielded errors close to $0.8$ which suggests that finite sample effects did not matter for the error).

Although one-dimensional confounding heavily violates our model assumptions, the results of both real data experiments look somehow positive.

## 5    Causal learning theory

So far, we have supported causal regularization mainly via transferring Bayesian arguments for regularization from scenario 1 to scenario 2. An alternative perspective on regularization is provided by statistical learning theory [2]. Generalization bounds guarantee that the *expected* error is unlikely to significantly exceed the *empirical* error for any regression function $f$ from a not too rich class $\mathcal{F}$. If $L(Y, f(X))$ denotes some loss function, they guarantee, for instance, that the following inequality holds with a certain probability uniformly for $f \in \mathcal{F}$:

$$\mathbf{E}[L(Y, f(\mathbf{X})] \leq \frac{1}{n} \sum_{j=1}^{n} L(y_i, f(\mathbf{x}_i)) + C(\mathcal{F}),$$

where $C(\mathcal{F})$ is some 'capacity term'.

In the same way, as these bounds relate empirical loss with expected loss, we will relate the expected (*statistical*) loss above with the *interventional loss*

$$\mathbf{E}_{do(\mathbf{X})}[L(Y, f(\mathbf{X})] := \int L(y, f(\mathbf{x})) p(y|do(\mathbf{x})) p(\mathbf{x}) d\mathbf{x}, \tag{14}$$

(which quantifies how well $f$ describes the change of $Y$ for interventions on $\mathbf{X}$) via a causal generalization bound of the form

$$\mathbf{E}_{do(\mathbf{X})}[L(Y, f(X)] \leq \mathbf{E}[L(Y, f(X))] + C(\mathcal{F}),$$

for some capacity term $C(\mathcal{F})$. Note that the type of causal learning learning theory developed here should not be confused with [23], which considers the generalization error of classifiers that infer cause-effect directions after being trained with *multiple* data sets of cause-effect pairs.

Figure 4 shows our confounding model that significantly generalizes our previous models. $\mathbf{Z}$ and $\mathbf{X}$ are arbitrary random variables of dimensions $\ell$ and $d$, respectively. Apart from the graphical structure, we only add the parametric assumption that the influence of $\mathbf{Z}$ on $\mathbf{Y}$ is linear additive:

$$Y = Y' + \mathbf{Z}\mathbf{c}, \tag{15}$$

where $\mathbf{c} \in \mathbb{R}^{\ell}$. The change of $Y$ caused by setting $\mathbf{X}$ to $\mathbf{x}$ via interventions is given by Pearl's backdoor criterion [5] as

$$p(y|do(\mathbf{x})) = \int p(y|\mathbf{x}, \mathbf{z}) p(\mathbf{z}) d\mathbf{z}. \tag{16}$$

Note that the observational conditional $p(y|\mathbf{x})$ would be given by replacing $p(\mathbf{z})$ with $p(\mathbf{z}|\mathbf{x})$ in (16). Interventional conditionals destroy the dependences between the confounder $\mathbf{Z}$ and the 'treatment' variable $\mathbf{X}$ by definition of an intervention. The supplement shows that the difference between interventional and observational loss can be concisely phrased in terms of covariances if we choose the loss $L(Y, f(X)) = (Y - f(X))^2$:

**Lemma 1** (interventional minus observational loss)**.** *Let* $g(\mathbf{x}) := \mathbf{E}[Y'|\mathbf{x}]$. *Then*

$$\mathbf{E}_{do(\mathbf{X})}[(Y - f(\mathbf{X}))^2] - \mathbf{E}[(Y - f(\mathbf{X}))^2] = (\Sigma_{(f-g)(\mathbf{X})\mathbf{z}})\mathbf{c}.$$

For every single $f$, the vector $\Sigma_{(f-g)(\mathbf{X})\mathbf{z}}$ is likely to be almost orthogonal to $\mathbf{c}$ if $\mathbf{c}$ is randomly drawn from a rotation invariant distribution in $\mathbb{R}^\ell$. In order to derive statements of this kind that hold *uniformly* for all functions from a function class $\mathcal{F}$ we introduce the following concept quantifying the capacity of $\mathcal{F}$:

**Definition 1** (correlation dimension). *Let $\mathcal{F}$ be some class of functions $f : \mathbb{R}^d \to \mathbb{R}$. Given the distribution $P_{\mathbf{X},\mathbf{Z}}$, the correlation dimension $d_{\mathrm{corr}}$ of $\mathcal{F}$ is the dimension of the span of*

$$\{\Sigma_{f(\mathbf{X})\mathbf{Z}} \mid f \in \mathcal{F}\}.$$

To intuitively understand this concept it is instructive to consider the following immediate bounds:

**Lemma 2** (bounds on correlation dimension). *The correlation dimension of $\mathcal{F}$ is bounded from above by the dimension of the span of $\mathcal{F}$. Moreover, if $\mathcal{F}$ consists of linear functions, another upper bound is given by the rank of $\Sigma_{\mathbf{XZ}}$.*

In the supplement we show:

**Theorem 2** (causal generalization bound). *Given the causal structure in Figure 4, where $\mathbf{Z}$ is $\ell$-dimensional with covariance matrix $\Sigma_{\mathbf{ZZ}} = \mathbf{I}$, influencing $\mathbf{X}$ in an arbitrary way. Let the influence of $\mathbf{Z}$ on $Y$ be given by a 'random linear combination' of $\mathbf{Z}$ with variance $V$. Explicitly,*

$$Y' \mapsto Y = Y' + \mathbf{Z}\mathbf{c},$$

*where $\mathbf{c} \in \mathbb{R}^\ell$ is randomly drawn from the sphere of radius $\sqrt{V}$ according to the Haar measure of $O(\ell)$. Let $\mathcal{F}$ have correlation dimension $d_{\mathrm{corr}}$ and satisfy the bound $\|(f - g)(\mathbf{X})\|_{\mathcal{H}} \le b$ for all $f \in \mathcal{F}$ (where $g(\mathbf{x}) := \mathbf{E}[Y'|\mathbf{x}]$). Then, for any $\beta > 1$,*

$$\mathbf{E}_{do(\mathbf{X})}[(Y - f(\mathbf{X}))^2] \le \mathbf{E}[(Y - f(\mathbf{X}))^2] + b \cdot \sqrt{V \cdot \beta \cdot \frac{d_{\mathrm{corr}} + 1}{\ell}},$$

*holds uniformly for all $f \in \mathcal{F}$ with probability $1 - e^{n(1-\beta+\ln\beta)/2}$.*

Note that $\Sigma_{\mathbf{ZZ}} = \mathbf{I}$ can always be achieved by the 'whitening' transformation $\mathbf{Z} \mapsto (\Sigma_{\mathbf{ZZ}})^{-1/2}\mathbf{Z}$. Normalization is convenient just because it enables a simple way to define a 'random linear combination of $\mathbf{Z}$ with variance $V$', which would be cumbersome to define otherwise.

Theorem 2 basically says that the interventional loss is with high probability close to the expected observational loss whenever the number of sources significantly exceeds the correlation dimension. Note that the confounding effect can nevertheless be large, that is, it would heavily spoil ordinary least square (e.g. unregularized) regression. Consider, for instance, the case where $\ell = d$ and $\mathbf{X}$ and $\mathbf{Z}$ are related by $\mathbf{X} = \mathbf{Z}$. Let, moreover, $Y' = \mathbf{X}\mathbf{a}$ for some $\mathbf{a} \in \mathbb{R}^d$. Then the confounding can have significant impact on the correlations between $Y$ and $\mathbf{X}$ due to $Y = \mathbf{X}(\mathbf{a} + \mathbf{c})$, whenever $\mathbf{c}$ is large compared to $\mathbf{a}$. However, whenever $\mathcal{F}$ has low correlation dimension, the selection of the function $f$ that optimally fits observational data is not significantly perturbed by the term $\mathbf{X}\mathbf{c}$. This is because $\mathbf{X}\mathbf{c}$ 'looks like random noise' since $\mathcal{F}$ contains no function that is able to account for 'such a complex correlation'. For the simple case where $\Sigma_{\mathbf{XZ}}$ has low rank, for instance, the term $\mathbf{Z}\mathbf{c}$ almost behaves like noise for typical $\mathbf{c}$ (w.r.t. any class $\mathcal{F}$ of linear functions), because the majority of components of $\mathbf{Z}$ are uncorrelated with $\mathbf{X}$, after appropriate basis change.

Since $\ell, d_{\mathrm{corr}}, b$ in Theorem 2 are unobserved, its value will mostly consist in qualitative insights rather than providing quantitative bounds of practical use.

# 6 What do we learn for the general case?

Despite all concerns against our 'hand-tuned' confounder model, we want to stimulate a general discussion about recommending stronger regularization than criteria of *statistical predictability* suggest, whenever one is actually interested in causal models. Our theoretical results suggest that this helps in particular when a type of confounding is expected that – if present – generates complex dependences, which would strongly regularized regression treat as noise. The advice of limiting the complexity of models to capture some causal truth could also be relevant for modern deep learning, since the goal of interpretability of algorithms for classification or other standard tasks could possibly be improved by having causal features rather than purely predictive ones.

It is, however, by no means intended to suggest that this simple recommendation would *solve* any of the hard problems in causal inference.

## Footnotes

[1]often for $d = 1$ and with a binary treatment variable $X$

[2]After submission I became aware of a preprint with the same title as mine, [11] where regularizers are constructed that are tailored particularly for causal features.

[3]Here we have, for simplicity, assumed $n > d$.

[4]Readers not familiar with operator theory may read all our operators as matrices with huge $n$ without loosing any essential insights – except for the cost of having to interpret all equalities as *approximate* equalities. To facilitate this way of reading, we will use $(\cdot)^T$ also for the adjoint of operators in $\mathcal{H}$ although $(\cdot)^*$ or $(\cdot)^\dagger$ is common.

[5]$E \sim \mathcal{N}(0, \sigma_E^2)$ and thus $e_j \sim \mathcal{N}(0, 1/n)$, which implies $\hat{E} \sim \mathcal{N}(0, \mathbf{I}/n)$ and thus $\widehat{\Sigma_{\mathbf{X}E}} = \hat{\mathbf{X}}^T \hat{E} \sim \mathcal{N}(0, \hat{\mathbf{X}}^T \hat{\mathbf{X}} \sigma_E^2 / n) = \mathcal{N}(0, \widehat{\Sigma_{\mathbf{X}\mathbf{X}}} \sigma_E^2 / n)$.

[6][17] constructs confounders for linear non-Gaussian models and [18] infer confounders of univariate $X, Y$ subject to the additive noise assumption.

[7][20] claims, for instance, "If $\ell_2$ was the norm of the 20th century, then $\ell_1$ is the norm of the 21st century ... OK, maybe that statement is a bit dramatic, but at least so far, there's been a frenzy of research involving the $\ell_1$ norm and its sparsity-inducing properties...."

[8]See also [21] for regularization by early stopping in a different context.

[9]In order to avoid overfitting issues, we decided in Ref. [14] to only generate low-dimensional data with $d$ around 10.

[10]Note that [14] also used normalization to achieve reasonable estimates of confounding for this case.

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
