[Supplementary Material · paper_neurips_2019_supplement.pdf]

# Supplement for "Causal Regularization"

**Dominik Janzing**
Amazon Research Tübingen
Germany
janzind@amazon.com

## 0.1 Proof of equation (8)

Due to $\Sigma_{\mathbf{XX}} = \mathbf{X}^T\mathbf{X}$ we have
$$\Sigma_{\mathbf{XX}}X^{-1} = X^{\dagger}XX^{-1} = X^{\dagger},$$
since $XX^{-1}$ is the orthogonal projection onto the image of $X$, which is orthogonal to the kernel of $X^T$. Then invertibility of $\Sigma_{\mathbf{XX}}$ implies
$$X^{-1}E = \Sigma_{\mathbf{XX}}^{-1}X^T E = \Sigma_{\mathbf{XX}}^{-1}\Sigma_{\mathbf{X}E}.$$

## 1 Rewriting Ridge and Lasso in terms of empirical covariance matrices

We first write $\hat{Y} = \hat{Y}_{\hat{\mathbf{X}}} + \hat{Y}_{\perp}$ where $\hat{Y}_{\hat{\mathbf{X}}}$ and $\hat{Y}_{\perp}$ denote the projections of $\hat{Y}$ onto the image of $\hat{\mathbf{X}}$ and its orthogonal complement, respectively. Then we can rewrite the empirical error as

$$\|\hat{Y} - \hat{\mathbf{X}}\mathbf{a}'\|^2 = \|\hat{Y}_{\hat{\mathbf{X}}} - \hat{\mathbf{X}}\mathbf{a}'\|^2 + \|\hat{Y}_{\perp}\|^2 = (\mathbf{a}' - \widehat{\Sigma_{\mathbf{XX}}}^{-1}\widehat{\Sigma_{\mathbf{XY}}})^T\widehat{\Sigma_{\mathbf{XX}}}(\mathbf{a}' - \widehat{\Sigma_{\mathbf{XX}}}^{-1}\widehat{\Sigma_{\mathbf{XY}}}) + \|\hat{Y}_{\perp}\|^2.$$

The second term does not depend on $\mathbf{a}'$ and is thus irrelevant for the optimization.

## 2 On the difficulty of mixing scenarios 1 and 2

Let us consider finite sample issues for scenario 2 in the purely confounded regime $\mathbf{a} = 0$. Then, $Y = \mathbf{Z}\mathbf{c}$ and the empirical correlations between $\mathbf{X}$ and $Y$ read

$$\widehat{\Sigma_{\mathbf{XY}}} = \widehat{\Sigma_{\mathbf{X}Z}}\mathbf{c} = M^T\widehat{\Sigma_{\mathbf{ZZ}}}\mathbf{c}. \tag{1}$$

Assuming that $\mathbf{c}$ is distributed according to an isotropic Gaussian $\mathcal{N}(0, \sigma_c^2\mathbf{I})$ for some parameter $\sigma_c$ (to resemble the distribution of $\hat{E}$ in scenario 1), the random vector (1) follows the distribution

$$\mathcal{N}(0, \sigma_c^2 M^T\widehat{\Sigma_{\mathbf{ZZ}}}^2 M), \tag{2}$$

if $\widehat{\Sigma_{\mathbf{ZZ}}}$ and $M$ are fixed. In the finite sample regime, $\sigma_c^2 M^T\widehat{\Sigma_{\mathbf{ZZ}}}^2 M$ is not a multiple of $\widehat{\Sigma_{\mathbf{XX}}} = M^T\widehat{\Sigma_{\mathbf{ZZ}}}M$, because $\widehat{\Sigma_{\mathbf{ZZ}}}$ is the identity only in the population limit. Hence, there is no simple relation between the distribution of $\widehat{\Sigma_{\mathbf{XY}}}$ and the matrix $\widehat{\Sigma_{\mathbf{XX}}}$, which has been crucial for our analysis of scenarios 1 and 2. For high dimensions $d$ and $\ell$ and random matrices $M$, one could possibly derive statements on the asymptotic relation between $M^T\widehat{\Sigma_{\mathbf{ZZ}}}^2 M$ and $M^T\widehat{\Sigma_{\mathbf{ZZ}}}M$ regarding spectra and spectral subspaces using free probability theory [1, 2].

## 3 Proof of Lemma 1

By definition, The difference between the two losses can be written as:

$$\int (y - f(\mathbf{x}))^2[p(y|\mathbf{x}) - p(y|do(\mathbf{x}))]p(\mathbf{x})d\mathbf{x} = \int (y - f(\mathbf{x}))^2 p(y|\mathbf{x}, \mathbf{z})\{p(\mathbf{x}, \mathbf{z}) - p(\mathbf{x})p(\mathbf{z})\}d\mathbf{z}d\mathbf{x}$$

$$= \mathbf{E}[(Y - f(\mathbf{X}))^2|\mathbf{x}, \mathbf{z}]\{p(\mathbf{x}, \mathbf{z}) - p(\mathbf{x})p(\mathbf{z})\}d\mathbf{z}d\mathbf{x}.$$

We rewrite the conditional expectation as

$$\mathbf{E}[(Y - f(\mathbf{X}))^2|\mathbf{x}, \mathbf{z}] = \mathbf{E}[(Y' + \mathbf{zc} - f(\mathbf{x}))^2|\mathbf{x}, \mathbf{z}]$$
$$= \mathbf{E}[Y'^2|\mathbf{x}, \mathbf{z}] + (\mathbf{zc})^2 + f(\mathbf{x})^2 + \mathbf{E}[Y'|\mathbf{x}, \mathbf{z}]\mathbf{zc} - \mathbf{E}[Y'|\mathbf{x}, \mathbf{z}]f(\mathbf{x}) - f(\mathbf{x})\mathbf{zc}.$$
$$= \mathbf{E}[Y'^2|\mathbf{x}] + (\mathbf{zc})^2 + f(\mathbf{x})^2 + g(\mathbf{x})\mathbf{zc} - g(\mathbf{x})f(\mathbf{x}) - f(\mathbf{x})\mathbf{zc},$$

where the last step used $Y' \perp\!\!\!\perp \mathbf{Z}|\mathbf{X}$ which follows from d-separation in the DAG in Figure 4. Since the above conditional expectation is integrated over $p(\mathbf{x}, \mathbf{z}) - p(\mathbf{x})p(\mathbf{z})$, only terms matter that contain both $\mathbf{x}$ and $\mathbf{z}$. We therefore obtain

$$\mathbf{E}[(Y - f(\mathbf{X}))^2] - \mathbf{E}_{do(\mathbf{X})}[(Y - f(\mathbf{X}))^2] = \int (g(\mathbf{x}) - f(\mathbf{x}))\mathbf{zc}\{p(\mathbf{x}, \mathbf{z}) - p(\mathbf{x})p(\mathbf{z})\}d\mathbf{z}d\mathbf{x}$$
$$= (\Sigma_{(g-f)(\mathbf{X}),\mathbf{z}})\mathbf{c}.$$

## 4  Proof of Theorem 2

We first need the following result which is basically Lemma 2.2 in [3] together with the remarks preceding 2.2:

**Lemma** [Johnson-Linderstrauss type result] *Let $P$ be the orthogonal projection onto an $n$-dimensional subspace of $\mathbb{R}^m$ and $v \in \mathbb{R}^m$ be randomly drawn from the uniform distribution on the unit sphere. Then $\|Pv\|^2 \geq \beta n/m$ with probability at most $e^{n(1-\beta+\ln\beta)/2}$.*

We are now able to prove Theorem 2. Let $\mathbf{c}^{\mathcal{F}}$ be the orthogonal projection of $\mathbf{c}$ onto the span of $\{\Sigma_{(g-f)(\mathbf{X})\mathbf{z}}|f \in \mathcal{F}\}$ (whose dimension is at most $d_{\text{corr}} + 1$). Note that the vector $\Sigma_{(g-f)(\mathbf{X})\mathbf{z}} \in \mathbb{R}^{\ell}$ has the components $\langle (g - f)(\mathbf{X}), Z_j \rangle$ if $Z_j$ denotes the components of $\mathbf{Z}$, which are orthonormal in $\mathcal{H}$. Hence

$$\|\Sigma_{(g-f)(\mathbf{X})\mathbf{z}}\| \leq b.$$

Thus the absolute value of the difference of the losses is bounded by

$$|\Sigma_{(g-f)(\mathbf{X})\mathbf{z}}\mathbf{c}^{\mathcal{F}}| \leq b\sqrt{V}\|\mathbf{c}^{\mathcal{F}}\|.$$

Then the proof follows from

$$\|\mathbf{c}^{\mathcal{F}}\| \leq \sqrt{\beta\frac{d_{\text{corr}} + 1}{\ell}},$$

due to the above Lemma.