[Reviews · NeurIPS 2019]

Reviewer 1



Reasons for score: ---------------------- Lack of clarity regarding some of the main theoretical and empirical results (see detailed comments and improvements for details). Assuming the authors address these points of clarification, my main concern is that the analyses that the authors present does not provide a practical method that practitioners can use: if I am understanding correctly, the conclusion is regularization might somewhat reduce the effects of confounding. But the authors do not provide a way to do sensitivity analysis to check how much confounding still exists or what to do about it; or what assumptions are required for their method to completely identify the causal estimands. Detailed comments: ------------------ Regarding the theory: - Some of my confusion arises from the fact that I do not fully understand what the authors mean by a "mixing matrix" and \ell "sources". I assumed that it is a random matrix based on their experimental setup where is drawn from a gaussian distribution. Clarification regarding those two terms would be helpful. - The authors outline a specific generative model under which the confounded case would coincide with the overfitting/finite sample OLS case. In that case, the unobserved common cause (Z) is drawn from a random distribution with mean 0, the effect that Z has on Y (c) is also drawn from a random distribution with mean 0, the effect that the common cause has on X (the mixing matrix) is also drawn from a random distribution with mean 0 and finally, the causal effect under investigation is also drawn from a random distribution. If I am not mistaken, this scenario describes a degenerate case where there is no confounding (and maybe less importantly the causal effect is assumed to be 0). In that case there is no need for confounding adjustment in the asymptotic regime because the expected value of Zc = 0 as n --> \infty. Can the authors clarify how this summary is a mischaracterization of their work? - Assuming that I am incorrect about the previous point, and confounding can occur in this setting. I do not believe that Theorem 1 implies identification of the causal estimands (without additional strong assumptions). Specifically, for regularization to produce asymptotically consistent estimates of a in the non-causal regularization case, the matrix X has to obey certain properties (roughly speaking, some version of the irrepresentability condition; more detailed discussion can be found in the reference in Lasso-type recovery of sparse representations by Meinshausen and Yu, 2009). The authors do not make any restrictions about the X vector which is necessary to ensure that a converges to something meaningful. - There is a bit of a discordance across some of the theoretical results: for the majority of the paper (theorem 1 and the experimental results), the authors focus on recovering a in the asymptotic regime as their main goal. In section 5, however, they switch gears to talk about the finite sample efficiency of the Y estimate. This switch is a bit confusing, since one does not imply the other: an asymptotically consistent estimate of a does not imply an efficient estimate of Y in finite samples. Would the authors be able to clarify what is the intended connection here? - One paradoxical finding that the authors arrive at is that when X=Z, the "deconfounding" task becomes harder. This is a bit surprizing since this discribes the case where there is no hidden confounding (all the unobserved confounders are observed if X is observed). Can the authors comment on that? Am I misunderstanding seomthing here? - This might be inconsequential for the authors' findings, but I believe the statement on line 139 about the distribution unregularized regression vector is not accurate in the general case. This is a case of omitted variable bias, which is known to be asymptotically biased. E.g., see Greene chapter 8 for a detailed discussion of the limiting distribution in omitted variable case. Regarding the results: - To check my understanding of the results presented in figure 2, does the second plot imply that ridge regression with normal xvalidation is roughly the same as no regularization?

Reviewer 2



This paper investigates the links between regularization and causality from different perspectives. Based on novel theoretical results regarding this link, an algorithm is proposed that yields confounder-corrected (regularized) regression vectors. The algorithm is applied to both simulated and real data. Finally, learning theory is used to provide further insights and bounds on the difference between expected and interventional loss. Strengths: - The paper is very well-written and well-structured. - It investigates a very interesting question and provides novel theoretical results, a practical algorithm to use the insights for practise, and supports the theoretical results by empirical applications. - I couldn’t find any problems in the theoretical derivations and proofs. - Finally, the paper nicely puts the results into perspective and cautions which interpretations cannot be drawn from the results. Major comment: - From the descriptions in the paper, it is hard to understand all the details what is shown in Figures 2 and 3. For example, what is the unregularized RSE in Figure 2? What does each point represent in Figure 3 (and how many points are there)? Minor comments: - The abstract could be improved to better capture the interesting concepts and contributions of the paper. - While some references are cited that discuss the link between regularization and causal inference, there is a larger body of literature on this topic of which a few other studies should be cited and briefly discussed. - One half-sentence could be added in lines 93-98 to ease the understanding of the mathematical derivations. - There are a few minor typos, e.g. in lines 54/55: OLS not LOS - I would choose a different formulation in line 71 of “… is responsible for the failure of OLS regression”. - A slightly longer discussion of the finite-sample case would be interesting. - I suggest choosing different colors in Figure 2 (not red/green). - Figure 2 is barely readable, figure 3 requires strong zooming-in (not readable in print), and the legends as well as axes labels could be improved. AFTER AUTHORS’ RESPONSE: I have read the authors’ response, and stay with my initial assessment. For me, the main priority would continue to be the improvement of the description of the empirical results. The explanations in the authors’ response are helpful, and I think that it would be nice to have a brief comment on why combining finite sample with confounding is challenging. Regarding related work, regarding the space constraints, it would be sufficient in my opinion to limit the insertion and discussion of additional references to one sentence.

Reviewer 3



Summary: The authors investigate the degree to which regularization in linear models can actually lead to better causal models. They show that in the case of linear confounding there is an elegant connection between ridge regression and estimating the true causal effect vector for the observed covariates. By making a few approximations, they come up with a practical algorithm, similar in spirit to the adaptive lasso (i.e. fit OLS, then choose the right lambda penalty and refit) but for causal inference with ridge (though they also look into lasso as well, because why not?). The algorithm is shown to work well in simulations and has interesting results on real data. They conclude with an alternative look at the problem, through a learning theory lens. Review: This is an excellent paper. It was a joy to read. The questions raised and points made were thought-provoking and informative. I appreciated that the authors were clear on their contributions and that they were not overstating them. Rather, the paper does exactly what it aims to do: stimulate a conversation. I'll continue that conversation here. - Can anything be said in the high-dimensional (n << p) case? - Does the adjusted ridge penalty always end up being larger than the cross-validation chosen one? Or are there cases where CV actually chooses to over-regularize? The authors repeatedly claim that a high-level point here is that causal inference may require "stronger" regularization, which makes me think the former is the case. - Is there a connection to information criterion methods in the lasso case? Let's assume our model is not just linear, but linear and sparse. In this case, one would expect the lasso (or elastic net) would work better for the usual reasons. In the case of the lasso, it explicitly restricts the degrees of freedom of the model by inducing sparsity. A common approach in choosing lambda for the lasso is to use information criteria like BIC or AIC that tradeoff the fit with the degrees of freedom. - Should we expect nonlinear models to also require stronger regularization? I understand that theoretically these models are much harder to work with. I'm simply looking for some intuition here. Is the suspicion that the ridge result due to the linearity of the confounding and covariates, or is there something more fundamental going on here? Overall, I do not have any substantially negative things to say about this paper. There were several typos that I think the authors should hunt down in a re-read. And it would of course always be nice to have more experiments. However, I think as-is this paper is a nice contribution and deserves to be accepted.

[Author Response · NeurIPS 2019]

We thank the reviewers for their valuable comments and appreciate that all three reviewers acknowledge that we honestly
discuss limitations of directly applying our mainly theoretical insights (which two reviewers find stimulating).

**Reviewer 1**: "The authors do not provide a way [...] to check how much confounding still exists or what to do about it;
or what assumptions are required for their method to completely identify the causal estimands." This would indeed be
desirable but it's probably one of the hardest problems of causal inference (if confounders are unobserved!).

"In that case there is no need for confounding adjustment in the asymptotic regime because the expected value of
$\mathbf{Zc} = 0$." Note that the parameters $M$ and $\mathbf{c}$ are only drawn *once* for every data set. If $\mathbf{c}$ is drawn from a Gaussian
with high variance, it induces arbitrarily large confounding bias. Note, as an aside, that population Ridge and Lasso
maximize $p(\mathbf{a}|\Sigma_{\mathbf{XX}}, \Sigma_{\mathbf{X}Y})$ regardless of the distribution of $M$, it could even be any fixed matrix.

"There is a bit of a discordance across some of the theoretical results [...] In section 5, however, they switch gears to talk
about the finite sample efficiency of the $Y$ estimate." This may be a misunderstanding too. Section 5 is not about finite
sample effects. It just motivates our learning theory via the analogy between inferring the interventional loss from the
observations loss (our goal) to inferring the population loss from the empirical loss (the usual goal of statistical learning
theory).

"One paradoxical finding that the authors arrive at is that when $\mathbf{X} = \mathbf{Z}$, the "deconfounding" task becomes harder. This
is a bit surprizing since this describes the case where there is no hidden confounding (all the unobserved confounders
are observed if $X$ is observed)." The case $\mathbf{X} = \mathbf{Z}$ is not trivial for two reasons. First, we do not assume that the observer
*knows* the mixing matrix $M$. Second, even if he/she does, it is still unclear what part of the correlations are due to the
causal vector $\mathbf{a}$ and what part due to the confounding part $\mathbf{c}$ (the joint covariance matrix of $\mathbf{X}$ and $\mathbf{Z}$ is degenerate).

"... I believe the statement on line 139 about the distribution unregularized regression vector is not accurate in the
general case." Note that the statement relies on our set of assumptions, for which it should be true.

Yes, figure 2, 2nd from the left, shows that cross-validation is not better than no regularization because the bias is
dominated by confounding and finite sample effects are almost irrelevant.

Consistency statements were not in our focus. On the one hand, the tight analogy between both scenarios suggests
that those could be transferred across them without novel insights. On the other hand, we had to drop some further
theoretical results already to satisfy the space constraints. Our focus is the analogy.

**Reviewer 2:** We can briefly comment on *why* combining finite sample with confounding is theoretically challenging.
Regarding related work, we will try to also add a few words, although it's not clear what to cut out instead. If the
reviewers feel that some other explanations are too lengthy, we are open to suggestions. We will improve readability of
the figures and their explanations in the text and also the abstract.

Derivations 93-98: $\hat{E} \sim \mathcal{N}(0, \sigma_E^2 I)$ by assumption. Thus, $\widehat{\Sigma_{\mathbf{X}E}} = \hat{\mathbf{X}}^T \hat{E} \sim \mathcal{N}(0, \sigma^2 \hat{\mathbf{X}}^T \hat{\mathbf{X}}) = \mathcal{N}(0, \sigma_E^2 \widehat{\Sigma_{\mathbf{XX}}})$. For
scenario 2: $\mathbf{c} \sim \mathcal{N}(0, \sigma_c^2 I)$ and thus $\Sigma_{\mathbf{X}E} = M^T \mathbf{c} \sim \mathcal{N}(0, \sigma_c^2 M^T M) = \mathcal{N}(0, \sigma_c^2 \Sigma_{\mathbf{XX}})$. Every point in figure 2
represents one of the 100 runs. Unregularized RSE means RSE with OLS estimator (which coincides with $\beta$ in the
population limit).

**Reviewer 3:** "Does the adjusted ridge penalty always end up being larger than the cross-validation chosen one?... "
Interesting question. For strong confounding, CV certainly under-regularized for causal purposes. In the regime where
the bias of $\mathbf{a}$ is dominated by finite sampling we would rather trust CV instead of claiming that it over-regularizes
just because our adjusted penalty regularizes less. After all, estimation of $\beta$ remains a bottleneck. We agree with the
intuition that sparse models behave even better for Lasso, and some preliminary experiments confirmed that intuition,
but we didn't elaborate on it. We also thought about using information criteria for selecting $\lambda$, but for scenario 2 this
would require the unobserved number $\ell$. We also derived statements on selecting $\lambda$ by cross-validation across *different*
*environments*, but we had to drop that due to lack of space.

"Should we expect nonlinear models to also require stronger regularization?" We believe so. The following high
level view may help: (1) Our influence of the confounder depends on multiple independently drawn parameters (the
entries of $\mathbf{c}$). (2) Due to a concentration of measure effect, the regression loss is likely to be close to its average over
the distribution of parameters (all $\mathbf{c}$) – uniformly over $\mathcal{F}$ if $\mathcal{F}$ is 'small'. (3) In our case, the average loss over all
distribution of parameters coincides with the interventional loss. Thus, the loss for a typical draw of $\mathbf{c}$ is close to the
interventional loss. Nonlinear models satisfying (1) can easily satisfy (2) if the regression loss depends *weakly* on each
of the confounding parameters. Then, Efron-Stein entails the same concentration of measure. How to achieve (3) in
nonlinear models is less obvious, however.

We have assumed $n > d$ just to get a concise description (e.g. unique minimum in (2)), the analogy between scenario 1
and 2 still holds for $n < d$ and $\ell < d$.

[Meta-Review · NeurIPS 2019]

This paper discusses the connection between regularization and causality, resting on the simple problem of linear regression, using ridge regression and Lasso as illustrative cases study for their argument. The paper provides original insights on the link between both regularization and causality. For the final version, it would be nice if the authors could introduce a bit more context on do-calculation (two lines stating that this is a pivotal tool from the framework of causality) and give more practical insights on the consequences of their results.